# Agricultural Land-Use Changes in the Judean Region from the End of the Ottoman Empire to the End of the British Mandate: A Spatial Analysis

**Gad Schaffer**

Multidisciplinary Studies Department, Tel-Hai Academic College, Upper Galilee 1220800, Israel;
schaffergad@telhai.ac.il; Tel.: +972-55-2236800

**Abstract:** Vines and olives are two important and widespread traditional agricultural crops that are also connected to the Judeo–Christian–Muslim tradition. The goal of the research was to demonstrate the importance of using cartographical sources to obtain a more accurate and complete view of the past. To this end, the aims were: (1) to reconstruct the former agricultural land-use in three periods, 1873–1874, 1917, and 1943–1945; (2) to analyze the different spatial physical factors that could explain the spatial distribution of traditional agricultural landscapes; (3) to identify the changes which took place between the three reconstructed timestamps. The research employed different cartographic sources and the implemented analyses were conducted using GIS tools and methods. The results show that, in the past, the distribution of vines and olive groves greatly depended on several physical geographic factors (climate, slopes, direction). Nonetheless, human factors such as political instability, cultural and religious beliefs contributed as well. Moreover, this research showed how GIS has advanced historical geography research. Lastly, the research demonstrated that obtaining the most complete view of the past can be achieved by a combination of sources together with the use of GIS tools and methods.

**Keywords:** land use/land cover (LULC); landscapes; historical maps; Geographic Information System (GIS); agriculture; vineyards; olive groves; Ein Karem; Bethlehem; Hebron

## 1. Introduction

The interdisciplinary field of Historical Geographic Information Systems (HGIS) was formed towards the end of the 1990s [1]. The novelty of HGIS lies in the use of Geographic Information System (GIS) software programs for the study of history, and later on for geography as well [2–4]. GIS software programs allow for the extraction, analysis, and synthesis of spatial components found in cartographic sources, a task that was previously almost impossible or very time-consuming [2]. The introduction of GIS to the field of History and Geography has increased the use of historical cartographical sources (maps, drawings, photographs, and aerial photos) for research [5,6].

In the last two decades, much research has been performed on the reconstruction of past landscapes and the examination of changes in land use/land cover (LULC) over time [7–10]. LULC research has increased awareness of LULC changes and their links to current environmental changes [11]. Moreover, landscapes comprise cultural meanings, values, beliefs, and ideologies, making each one of them unique [12,13]. Landscapes, or more precisely, cultural landscapes, were recognized by UNESCO in 1992 as important "combined works of nature and man" to be acknowledged and protected [14] (p. 18). In 2014, UNESCO declared the Battir cultural landscape as a World Heritage Site. Battir is a small Palestinian village in the West Bank, located south-west of Jerusalem, in the Judean region (Figure 1). The nomination of Battir as a World Heritage Site was due to its special farmed valleys composed of old stone terraces, which was the traditional way of irrigating

agricultural land in this part of the world. The title that UNESCO awarded to the Battir Cultural Landscape was Land of Olives and Vines [15].

Following the agricultural revolution about 10,000 years ago, humans started to settle down where good physical conditions such as fertile soil, water, and moderate temperatures could increase the production of good and abundant crops, thus providing plenty of food [16]. Gradually, agricultural crops became important for the survival of the increased population [16]. Agriculture was also important since its products could be traded for other products or for money. The agricultural crops in each geographical area were different, especially because of the unique physical conditions of each region. According to various archaeological findings and written historical sources, the main crops in Palestine were wheat, barley, legumes, and grapes, as well as fruit trees, mainly olive, date and pomegranate [17]. Often, these crops were given additional cultural symbolism and religious importance. This was the case with the olives and vines [18].

In many parts of the Mediterranean region and in the eastern regions of the Middle East, vineyards and olive groves have been a part of the landscape for centuries [19–21]. Olive trees originated in the Mediterranean region and were domesticated around 6000–7000 years ago [22]. Vines (Vitis vinifera) are climbing plants that originated in different places around the world [17]. Viticulture as a source of wine has always been one of the most important and widespread agricultural land-uses due to its significant presence in the Judeo–Christian tradition [23,24]. Likewise, the olive tree and its products are deeply rooted in the Judeo–Christian–Muslim tradition [25]. In the Bible, vines are mentioned hundreds of times, reflecting their substantial role in agriculture. Furthermore, the names of many historical settlements in Palestine derived from the word vines [24]. For example, *Gat*, which in Hebrew means a winepress device, and *Ein Karem*, which in Hebrew means the spring of vineyards. While olives and vines were important crops in Palestine, the number of vines and olives fluctuated throughout history. Some causes of these fluctuations were climate-related, for example, years of drought destroyed vine plantations [26,27]. Another reason was that throughout history, various pests attacked and damaged crops, such as the grape phylloxera insect, which attacks vines [23]. Other reasons were anthropogenic, such as political instability generated by conflicts and wars in the area that caused the locals to migrate and abandon their crops [25]. Additionally, the fluctuations were connected to trends in chain supply and demand, which changed over the years. Changes in culture and religious beliefs also affected crops, such as the prohibition of wine production, to varying degrees, due to the application of Islamic laws in several periods [23,28]. For example, during the Muslim conquest (636–1099 AD), the Caliphate ordered the destruction of all wine-producing vineyards and only allowed the growth of a type of vineyard which produced table grapes [23]. Additionally, in the Ottoman period, the prohibition of growing vines for wine was mostly enforced until the middle of the nineteenth century [23].

At the beginning of the nineteenth century, the power of the Ottoman empire that ruled Palestine from 1516 began to diminish, and Palestine was left to fend for itself [29]. In that period, Palestine suffered from political instability, security problems, and a shrinking population [29,30]. With the continuing decline of the Ottoman empire towards the mid-nineteenth century, European interests in the region began to increase together with the beginning of Jewish immigration towards the end of the century. This was apparent from the capitulation treaties, which facilitated foreign investment, development, and exploration of Palestine [29,31]. The late nineteenth century is considered as the beginning of Palestine's modernization, which consisted of the construction of new buildings and infrastructures such as the Hejaz railroad, the start of foreign trade and a significant growth in population that enhanced further development [32]. During World War I, the Ottoman Empire collapsed, and Palestine came under the rule of the British Mandate for the next 31 years. Under the British Mandate, Palestine was classified as an agricultural land, as was the case for most British colonies [33]. During the last years of Ottoman rule, olive groves were present all across Palestine, but vineyards could only be found in two locations in the Galilee (around Kefr Birim, Sasa, and el Fish, and between Beit Jenn and Rameh), in a

single location in Samaria (around Mezrah esh Sherkiyeh), and in three locations in Judea (around Ein Karim, Bethlehem, and Hebron).

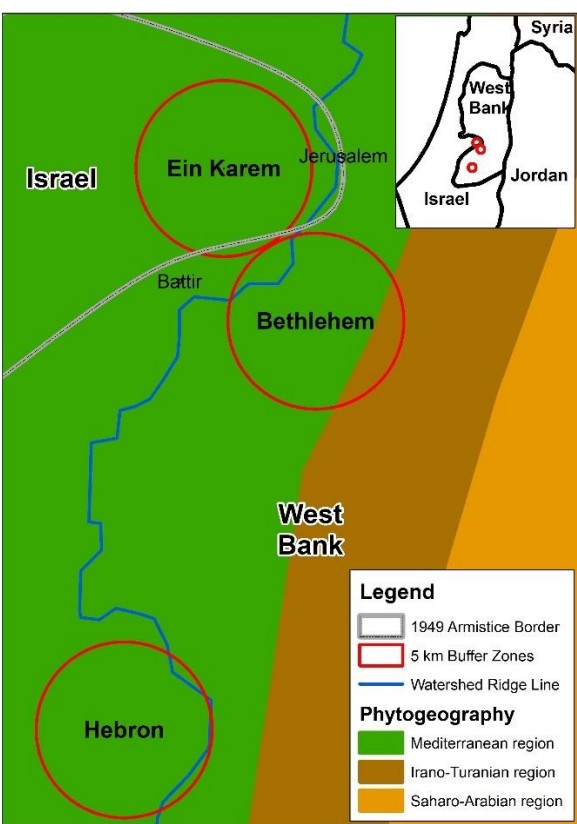

**Figure 1.** The three research areas—Ein Karem, Bethlehem, and Hebron. Each research area is circled in red, which delineates a 5 km radius from the main settlement outwards. The watershed ridge line of the Judaean and Samarian mountains and the phytogeographical regions are also shown (the phytogeographical regions were made on the basis of the 1984 Floristic Regions of the Land of Israel map [34]).

The goal of the research was to demonstrate the importance of using cartographical sources to obtain a more accurate and complete view of the past. The research spans 72 years that consist of over three timestamps, between 1873 and 1945. The reasons for choosing this period were twofold. First is the rapid change including substantial geo-political shifts. The second reason for ending the research period with 1945 was practically technical. Between 1948 and 1967, two of the research areas located in the West Bank were under Jordanian control. The research only found reprints of older maps with no new additions. Large-scale maps of other periods were not available, and small-scale maps did not represent the examined categories. Moreover, in the years since 1945, the categories found on the maps have changed, and in some maps the categories examined (i.e., vineyards and olive groves) became more general (i.e., agricultural area), which made it impossible to compare them with previous years.

To achieve this goal, the research examined the agricultural landscapes found in the Judean region, focusing on vineyards and olive groves. More specifically, three objectives were set towards achieving the goal of this research: (1) to reconstruct the former agricultural land-use over three periods, 1873–1874, 1917, and 1943–1945; (2) to analyze the different spatial physical factors that could explain the spatial distribution of agricultural landscapes; and (3) to identify the changes which took place between the three reconstructed timestamps.

## 2. Materials and Methods

### 2.1. Research Area

The geographic areas chosen for the research were three 5 km radius areas around three settlements in the Judean region—Ein Karem, Bethlehem, and Hebron (Figure 1). Each radius consisted of an area of 78.5 km$^2$. In the preliminary stage of the research, the 1881 Palestine Exploration Fund (PEF) map, which was the first source to be used in this research, was quickly scanned in order to identify areas which had both vineyards and orchards in the same location. Due to the presence of more areas in the Judea region than in the Galilee and Samaria, it was decided to focus on the three areas there, as the available sample was broader and could thus paint a clearer picture. The choice of a 5 km radius around these settlements was made because the distance from the center of these settlements to the boundaries of the agricultural crops found on the PEF map was approximately 5 km.

### 2.2. Research Materials

To reconstruct the changes in land-use over time, three sets of British maps were used. The first was the PEF map at a scale of 1:63,360, published in 1881 [35] (Table 1). The PEF map is considered the most accurate and precise map of nineteenth-century Palestine [36–38]. The survey was carried out between 1871–1878 by two British army officers, Claude Conder and Horatio Kitchener [39]. Specifically, the areas studied in this research were surveyed between 1873–1874. The second set of maps were British military maps at a scale of 1:25,000, published in 1917 (Table 1); these maps were copies of the original 1881 PEF map, with updated information. The third set of maps used were British Mandatory maps at a scale of 1:20,000, published between 1943–1945 (Table 1). Another source used was a world digital elevation model (DEM) at a resolution of 1-ARC (Table 1). A physiographic map of Israel from 1995 and a 1984 Floristic Region map of Israel were also used (Table 1) [34,40]. A written source that was widely used in this research was the Survey of Western Palestine: Memoirs of the Topography, Orography, Hydrography, and Archaeology—Judea [41]. The memoirs were written along with the PEF survey and describe the surveyed areas in great detail.

**Table 1.** The cartographical sources used in the research. For each cartographical source, the following details were provided: the agency that created the map, the name of the source, the year the source was published, the scale of the source and, if the source was georeferenced in this research, and the total Root Mean Square Error (RMSE).

| Mapping Agency | Name of Source | Year of Publication | Source Scale | Total RMSE in Meters |
|---|---|---|---|---|
| Palestine Exploration Fund | Survey of Western Palestine [35] | 1881 | 1:63,000 | Previously georeferenced |
| War Office, London | Sheet 17 Jerusalem [42] | 1917 | 1:63,360 | 27.4 |
| War Office, London | Sheet 21 Hebron [43] | 1917 | 1:63,360 | 38.8 |
| Survey of Palestine | Sheet 16/13 Ein Karim [44] | 1945 | 1:20,000 | 0.4 |
| Survey of Palestine | Sheet 17/12 Talpiyot [45] | 1943 | 1:20,000 | 5.6 |
| Survey of Palestine | Sheet 16/12 Bethlehem [46] | 1944 | 1:20,000 | 6.9 |
| Survey of Palestine | Sheet 16/11 Beit Fajjar [47] | 1943 | 1:20,000 | 6.2 |
| Survey of Palestine | Sheet 17/11 Tel Hordos [48] | 1943 | 1:20,000 | 2.0 |
| Survey of Palestine | Sheet 15/10 Hebron West [49] | 1945 | 1:20,000 | 4.3 |
| Survey of Palestine | Sheet 16/10 Hebron East [50] | 1944 | 1:20,000 | 3.9 |
| Survey of Israel | Physiographic Map of Israel [51] | 1995 | unknown | 572.9 |
| Israeli Ministry of Defense & Society for Protection of Nature | Floristic Regions of the Land of Israel [34] | 1984 | unknown | 181.2 |
| United States Geological Survey | World DEM SRTM1N31E035V3 [40] | 2014 | | Previously georeferenced |

### 2.3. Research Methods

The georeferencing, digitization, analysis, and examination of the different datasets were all performed using ArcGIS (10.5.1) software (Figure 2). The first step of this research was to geo-reference the cartographical sources. The GIS software allows us to examine the accuracy of the maps by comparing them to more accurate present-day sources. During the georeferencing process, the GIS software also calculates the Root Mean Squared Error (RMSE) [52]. The higher the RMSE number, the less spatially accurate the cartographical source. The 1881 PEF map, as well as the 2014 World DEM image, were already georeferenced and available in GCS WGS 1984, a world coordinate system [36,53]. The 1917 maps and the 1943–1945 maps were georeferenced using the graticule intersections (the coordinate grid points on the maps). The 1917 and 1943–1945 maps had a low RMSE, meaning they were quite accurate and could be used (Table 1). Since the coordinates in these sources were in the old local geographic coordinate system, the Israeli Cassini Soldner, the sources were first georeferenced to Israel's new geographic coordinate system, the Israeli Transverse Mercator, and were then projected into the world coordinate system, GCS WGS 1984, using the Project tool in ArcGIS. The physiographic map of Israel and the Floristic Regions of the Land of Israel were georeferenced using ten recognizable shared control points (i.e., Haifa Bay, Rosh HaNikra, Cape Costigan in the Dead Sea, etc.) found between the maps to an already georeferenced National Geographic map.

The second step of the research was to determine the agricultural categorization of the land-use. In the PEF and 1917 maps, the only two categories of agricultural land-use were vineyards and orchards. In the 1943–1945 maps, agricultural land-use was divided into five crop categories—vineyard, olive grove, banana grove, citrus grove, and orchards. In this research, the four land-use categories chosen for digitization were vineyards, olive groves, orchards, and built-up areas. The research focused on traditional agricultural crops used in this region, particularly vineyards and olive groves. For a better understanding of the settlement's linkage with local agricultural land-use, the built-up areas were also digitized. In the third step of the research, the land-use categories were digitized in all three time periods—1873–1874, 1917, and 1943–1945. The digitization was conducted on a scale of 1:5000–1:15,000.

The fourth step of the research was to compare and analyze the newly created digitized layers with different spatial factors. To quantitatively identify the changes in land-use categories over time, the Summarize tool was used in ArcGIS. In addition, the Tabulate Area (Spatial Analyst) tool was used to examine whether the land-use area categories of 1873–1874 remained the same in 1943–1945.

The fifth and last step was to examine whether physical spatial factors could explain the distribution and changes in land-use categories, and accordingly, the digitized layer was compared with elevation, slope, and aspect to the sun, using the 2014 world DEM. To compare the land-use categories to the elevation, the 2014 DEM raster was transferred into a polygon using the Raster to Polygon (Conversion) tool. Following that, the Intersect tool was used to perform a calculation between the land-use digitized layer and the polygon DEM layer. Finally, the compared data were extracted with a focus on the average area using the Summarize tool. To examine whether a link exists between the land-use layers and slopes, the 2014 DEM was used to measure the gradient of each area using the Slope (Spatial Analyst) tool. The new created slope layer was compared to the land-use layers using the Zonal Statistics as Table tool, from which data were extracted. Lastly, to examine in which direction the land-use categories found were facing with respect to the sun, the 2014 DEM was used to create a layer of directions to the sun using the Aspect (Spatial Analyst) tool. The created polygon layer was composed of ten different directions. To make the results easier to comprehend, the Aspect layer was first transformed into a raster using the Polygon to Raster (Conversion) tool. Later, using the Reclassify (Spatial Analyst) tool, the classes of the original Aspect layer were reclassified into four major directions: north, from 0°–45° and from 315°–0°, east from 45°–135°, south from 135°–225°, and west from 225°–315°. Once the reclassification was complete and a new raster was produced, it was

compared with the land-use layers using the Tabulate Area (Spatial Analyst) tool, from which the data were extracted to an Excel sheet.

| (1) Georeferencing |
|---|
| - historical maps from 1917 and 1943 – 1945 (using the graticule intersections) |
| - physiographic map of Israel and the Floristic Regions of the Land of Israel (using recognizable shared control points) |

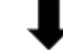

| (2) Examining the historical maps |
|---|
| - determining the land-use categories that will be digitized (orchards, vineyards, olive groves and built-up areas) |

| (3) Digitization of land-use categories |
|---|
| - digitizing the above land-use categories |

| (4) Examining land-use changes |
|---|
| - quantitatively identifying land-use changes in the three timestamps (Summarize) |
| - examining the correlation of land-use transformations between 1873 – 1874 and 1943 – 1945 (Tabulate Area) |

| (5) Examining links between physical factors and land-use (using the 2014 world DEM) |
|---|
| - Elevation (Raster to Polygon [Conversion] → Intersect tool → Summarize) |
| - Slope (Slope [Spatial Analyst] → Zonal Statistics as Table) |
| - Sun direction (Aspect [Spatial Analyst] → Polygon to Raster [Conversion] → Reclassify [Spatial Analyst] → Tabulate Area) |

**Figure 2.** Scheme showing the steps conducted in this research.

## 3. Results

### 3.1. Overall Changes in Land-Use Categories over Time

The agricultural land-use areas (orchards, vineyards, and olive groves) amount to 12–23% of the total research areas in 1873–1874, 10–23.5% in 1917, and 27.5–41% in 1945 (Table 2).

As shown in Table 2, in 1873–1874, Bethlehem had the largest area of orchards (20.73% of the total research area), followed by Ein Karem and Hebron. In contrast, Hebron had the largest area of vineyards (15.5% of the total research area), with Ein Karem and Bethlehem far behind. In 1917, as well as between 1873–1874, a similar trend was found, with Bethlehem with the largest area of orchards and Hebron with the largest area of vineyards. Between 1873–1874 and 1917, the results show a decrease in all rations of agricultural land areas with the exception of an increased percentage of total area of vineyards in Hebron (+18%). In 1943–1945, unlike previous examined years, a new separate land-use category emerged-olive groves—which, in both the 1873–1874 and 1917 maps, was included in the orchard category. Interestingly, in 1943–1945, Hebron still had the largest percentage of total area consisting of vineyards (26.11%) with an addition of orchards (21.91%). Between 1917 and 1943–1945, there was a noticeable increase in all three research areas in the total

area of all agricultural lands. As depicted in Figure 3, the dominant land-use in Ein Karem and Bethlehem was orchards, while in Hebron it was vineyards. Moreover, in Hebron to a larger extent, and in Bethlehem to a smaller extent, the orchards in the 1873–1874 and 1917 maps do not expand much to the east. While in Hebron in the 1943–1945 maps, the agricultural land has expanded across the entire research area and to the east, in Bethlehem the expansion of agricultural land was mostly to the north, west and south, leaving the east almost empty of this type of land.

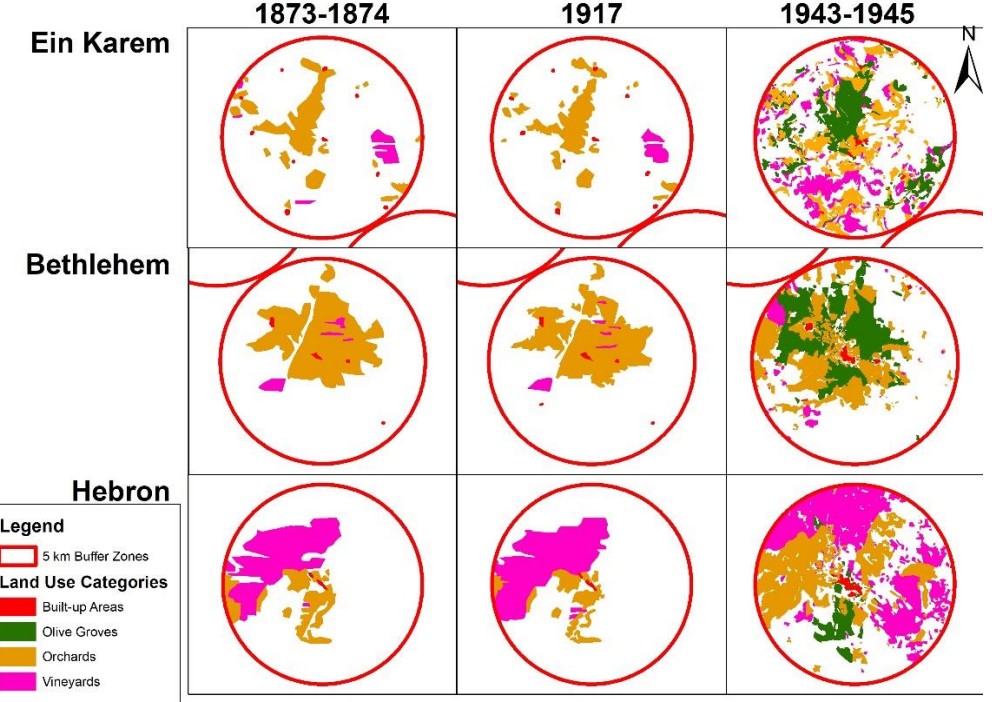

**Figure 3.** Changes in agricultural land-use in the research areas. The figure is divided into the three research areas, Ein Karem, Bethlehem and Hebron, and each one is divided into the three time periods, 1873–1874, 1917, and 1943–1945. Land uses are divided into four categories—built-up areas, olive groves, orchards, and vineyards.

**Table 2.** Agricultural land-use changes in the research areas (Ein Karem, Bethlehem, and Hebron) for each period examined, 1873–1874, 1917, and 1943–1945. The total area for each category is presented both in square kilometers and percentage of the total research area. Additionally, the dynamic percentage of land use changes in each category is presented for the periods between 1873–1874–1917 and 1917–1943–1945.

| Land-Use Categories | 1873–1874 | | 1873–1874–1917 | 1917 | | 1917–1943–1945 | 1943–1945 | |
|---|---|---|---|---|---|---|---|---|
| | Total Area (km²) | Total Area (%) | Dynamics of Change (%) | Total Area (km²) | Total Area (%) | Dynamics of Change (%) | Total Area (km²) | Total Area (%) |
| **Ein Karem** | | | | | | | | |
| Olive-Groves | n/a | n/a | n/a | n/a | n/a | n/a | 9.3 | 11.8 |
| Orchards | 7.8 | 10.0 | −10.4 | 7.0 | 8.9 | +48.6 | 10.4 | 13.3 |
| Vineyards | 1.5 | 2.0 | −21.9 | 1.2 | 1.5 | +549.1 | 7.9 | 10.0 |
| **Bethlehem** | | | | | | | | |
| Olive-Groves | n/a | n/a | n/a | n/a | n/a | n/a | 13.2 | 16.8 |
| Orchards | 16.3 | 20.7 | −9.1 | 14.8 | 18.8 | −1.9 | 14.5 | 18.5 |
| Vineyards | 1.1 | 1.4 | −20.9 | 0.9 | 1.1 | +93.5 | 1.7 | 2.1 |
| **Hebron** | | | | | | | | |
| Olive-Groves | n/a | n/a | n/a | n/a | n/a | n/a | 3.1 | 3.9 |
| Orchards | 5.7 | 7.2 | −25.2 | 4.2 | 5.4 | +305.5 | 17.2 | 21.9 |
| Vineyards | 12.2 | 15.5 | +18.0 | 14.4 | 18.3 | +42.7 | 20.5 | 26.1 |
| Total ResearchArea | 78.5 | | | 78.5 | | | 78.5 | |

An analysis of the continuity of area types found that only a small part of agricultural land in 1873–1874 was still agricultural in 1943–1945 (Table 3). For example, in Hebron in 1943–1945, only 2.9 km$^2$ of vineyards were located in the same area that had vineyards in 1873–1874. In Ein Karem and Bethlehem, the numbers were minimal (Table 3). It is interesting to note that many olive-grove areas in 1943–1945 occupy the same areas as orchards in 1873–1874. For example, the total amount of olive groves in Bethlehem in 1943–1945 was 13.2 km$^2$ (Table 2) and more than half of this area, 7.7 km$^2$, was found to be orchard areas in 1873–1874 (Table 3). This could mean that more than half of the previous orchards in Bethlehem were olive groves. The same might be true of Ein Karem and Hebron (Tables 2 and 3).

**Table 3.** The continuity of location conversion between the agricultural land-use categories in 1873–1874 with the land cover in 1943–1945. The table shows the amount of change in square kilometers.

| | 1873–1874 Land-Use Categories—Area in km$^2$ | | | | | |
| | Orchards | Vineyards | Orchards | Vineyards | Orchards | Vineyards |
| **1943–45 Land-Use Categories** | **Ein Karem** | | **Bethlehem** | | **Hebron** | |
|---|---|---|---|---|---|---|
| Olive Groves | 3.2 | 0.1 | 7.7 | 0.5 | 1.2 | 0.1 |
| Orchards | 1.1 | 0.1 | 4.4 | 0.5 | 1.8 | 7.1 |
| Vineyards | 0.6 | 0.1 | 0.0 | 0.0 | 0.1 | 2.9 |

### 3.2. Physical Factors Explaining Spatial Distribution of Land-Use Categories

#### 3.2.1. Elevation and Slope

Results of the correlation between land-cover categories, elevation, and slopes are presented in Table 4. With respect to elevation, in all the three research areas, vineyards were found on average in higher areas than orchards. For example, in Hebron, vineyards could be found at an elevation of 938–939 m while orchards were found at an elevation of 890–923 m. A similar trend can be seen in the other two research areas. With regard to olive groves, in Ein Karem, the average elevation of olive groves (624 m) in 1943–1945 resembled the average elevation of orchards (626 m), a difference of 2 m (Table 4).

The slope results show that the land-use categories could be found in areas between 26 and 35 degrees. In Hebron, the slope degree was higher than in Bethlehem and Ein Karem. Moreover, in Hebron, vineyards were found on higher slopes (31 degrees) than orchards (26–29 degrees) in 1873–1874 and 1917, unlike Ein Karem and Bethlehem. In 1943–1945, olive groves could be found across a large range of slopes (Table 4).

**Table 4.** The correlation between land-cover categories to elevation and slopes for each period examined.

| | Average Elevation in Meters | | | Average of Slope in Degrees | | |
| **Land-Use Categories** | **1873–1874** | **1917** | **1943–1945** | **1873–1874** | **1917** | **1943–1945** |
|---|---|---|---|---|---|---|
| | **Ein Karem** | | | | | |
| Olive-Groves | n/a | n/a | 624 | n/a | n/a | 31 |
| Orchards | 635 | 626 | 678 | 35 | 35 | 31 |
| Vineyards | 750 | 774 | 729 | 26 | 25 | 35 |
| | **Bethlehem** | | | | | |
| Olive-Groves | n/a | n/a | 729 | n/a | n/a | 32 |
| Orchards | 706 | 704 | 742 | 27 | 27 | 29 |
| Vineyards | 719 | 721 | 824 | 26 | 20 | 28 |
| | **Hebron** | | | | | |
| Olive-Groves | n/a | n/a | 885 | n/a | n/a | 18 |
| Orchards | 890 | 894 | 923 | 29 | 26 | 31 |
| Vineyards | 939 | 936 | 938 | 31 | 31 | 29 |

### 3.2.2. Aspect

The results of the *Aspect* examination are presented in Table 5. Most of the vineyards in Ein Karem and Bethlehem across all periods were facing eastwards, while in Hebron they mostly faced southwards. With regard to orchards, in Ein Karem and Hebron most of them faced southwards, while in Bethlehem they mostly faced eastwards. The olive-grove aspect strongly resembled that of orchards. Most olive groves in Ein Karem and Hebron were southwards-facing, like the orchards in those areas, and in Bethlehem most olive groves faced eastwards like the rest of the orchards in this area.

**Table 5.** The downhill direction in relation to the sun. Each research area, shown on the left side, is subdivided into one of the three periods (1873–1874, 1917, and 1943–1945) and then further subdivided into land-use categories (vineyards, orchards and olive groves). The upper columns preset the percentage out of the total area that shows the division into four downhill directions in relation to the sun—north, east, south, and west.

| Research Area | Year | Land-Use Category | % of the Total Area | | | |
| --- | --- | --- | --- | --- | --- | --- |
| | | | North | East | South | West |
| Ein Karem | 1873–1874 | Vineyards | 15.3 | 50.5 | 23.9 | 10.3 |
| | | Orchards | 26.8 | 19.8 | 36.8 | 16.6 |
| | 1917 | Vineyards | 13.6 | 55.7 | 21.4 | 9.2 |
| | | Orchards | 26.5 | 18.5 | 36.0 | 18.9 |
| | 1945 | Vineyards | 24.4 | 24.5 | 29.3 | 21.7 |
| | | Orchards | 31.2 | 22.8 | 26.7 | 19.3 |
| | | Olive Groves | 26.1 | 22.3 | 30.4 | 21.2 |
| Bethlehem | 1873–1874 | Vineyards | 32.3 | 33.3 | 21.5 | 12.8 |
| | | Orchards | 21.2 | 37.6 | 30.4 | 10.8 |
| | 1917 | Vineyards | 28.1 | 32.2 | 27.9 | 11.9 |
| | | Orchards | 22.6 | 37.9 | 28.6 | 10.9 |
| | 1945 | Vineyards | 28.4 | 29.1 | 24.3 | 18.2 |
| | | Orchards | 26.1 | 30.5 | 27.0 | 16.4 |
| | | Olive Groves | 23.8 | 38.2 | 29.6 | 8.5 |
| Hebron | 1873–1874 | Vineyards | 18.2 | 22.4 | 32.3 | 27.2 |
| | | Orchards | 21.9 | 23.9 | 36.0 | 18.2 |
| | 1917 | Vineyards | 17.9 | 21.8 | 32.8 | 27.6 |
| | | Orchards | 26.1 | 25.9 | 32.9 | 15.2 |
| | 1945 | Vineyards | 20.5 | 24.9 | 33.8 | 20.7 |
| | | Orchards | 22.1 | 22.6 | 34.1 | 21.2 |
| | | Olive Groves | 9.5 | 36.4 | 43.8 | 10.2 |

## 4. Discussion

### 4.1. Vineyards—Land-Use Change

According to the PEF map, in the period of the survey of Palestine between 1871–1878, there were only six large areas of vineyards. Two areas were found in the Galilee region, one in the Samaria region, and three in the Judean region. Indeed, in general terms these three regions have good conditions for growing vines [17,54–56]. These regions are located at a latitude of 31°, the average temperature is around 10–20 °C and usually does not drop below −4 °C nor exceed 30 °C. Additionally, the average annual rainfall is between 550 and 700 mm. Despite the good conditions in many places in Palestine, the Ottoman occupation and the Islamic ban on drinking alcohol led to a decline in wine production

and the vineyards that produced it [17,57]. Notwithstanding the ban on wine, it was possible to assume that vineyards would be found in cities where Jews and Christians Arabs lived, but that was not the case. In the Judea region at the end of the nineteenth century, Bethlehem had a Christian majority (the population was 5000 people of whom only 300 were Muslims) and Ein Karem was an all Christian village of 600 people, and yet the area size of the vineyards there was very small compared to the size of the vineyards in Hebron, which had a Muslim majority (the population of Hebron numbered 10,000 of whom 1000–12,000 were Jews) [41]. The reason vineyard areas were very extensive in Hebron, a predominantly Muslim settlement, was that these vineyards produced other non-alcoholic vine products [57]. Vines were grown for table grapes, raisins, and the production of a concentrated grape syrup [17]. With the slow decline of the Ottoman Empire in the late nineteenth century, there was a gradual increase in vine plantations for wine production. Initially, this was carried out mainly by Christian Arabs who, for religious reasons, were allowed to plant and produce wine for personal use and for ceremonies. Nonetheless, these plantations were small in size. The first extensive vineyards were planted by new Jewish settlers in Rishon LeZion in 1885, followed by other vineyards in Rosh Pina and Zikhron Ya'akov [23]. The new Jewish settlements brought vines for wine production from France. Nonetheless, this did not last long, since in 1890 vines were attacked by the grape phylloxera insect pest and many vines had to be uprooted [23,58]. During the First World War, other agricultural crops were more crucial to the war effort, and the vine industry declined further. It was only from 1936 on that vines started to be planted in increasing amounts [17]. The results of this research show that, during World War I (1917), there was a reduction of 9–25% in the total area of orchards and vineyards (Table 2).

This research illustrates that the largest area of vineyards in all three periods examined herein was in Hebron, as also described in the PEF memoirs: "the neighborhood of the Hebron hills is one of the principal vine-growing districts" [41] (p. 319). Hebron has better physical factors for growing vines than the other two examined areas. The annual rainfall average in Hebron is around 520–550 mm/year, its altitude (927 m) results in lower temperatures (yearly average temperature in winter is 7.5–10 °C and 22 °C in the summer). Interestingly, if we examine the soil type, it is the same in all three research areas—Terra Rossa and Rendzina [34]. Nonetheless, the PEF memoirs mention that Hebron's valleys "have good soil in them" [41] (p. 297). Ilan (1984) argues that, in the nineteenth century, most of the vineyards in the Hebron area were planted in good, deep, fertile Terra Rossa soil, which has good drainage [33]. Indeed, for vines, the most important aspect of soil is the draining of water [55]. Lastly, most vineyards in Hebron were found on south-facing slopes, a direction that is an advantage in the Northern Hemisphere, because it attracts the warmth of the afternoon sun, which improves ripeness and can protect against frost [55].

It is interesting to note that the PEF memoirs describe Hebron as "surrounded with fine vineyards" (p. 309) and maintain that the vineyards "extended over 6 miles", that is, 9.6 km. Although, as the 1873–1874 map shows, the vineyard areas were extensive, the study did not find vineyards which extended beyond 5.5 km from Hebron.

Another interesting point is that written sources tell us that, at the end of the nineteenth century, new vineyards were planted by Christian Arab and Jewish immigrants [23]. However, the second period examined in this research (1917) shows a strong decrease in vineyard areas in the research areas (except in Hebron). The reason for that might be that World War I had taken a toll on the vineyard industry.

If we take a look at the changes in vineyards throughout the years, the results show a marked increase in vineyards from 1917 to 1943–1945 in all three research areas, and especially in Hebron and Ein Karem. The first aid that the British Mandate gave to the agricultural sector was during its military regime (1917–1920), following the extensive damage caused to the agricultural sector in Palestine during World War I. A large amount of agricultural produce in Palestine was intended for export, and during the war there was a drastic decline in exports and many vineyards and orchard areas shrunk [17,57]. Under the direction of Colonel E.R. Sawer, the British decided on several actions, among them the

renewal of traditional agricultural crops, such as vines and olives. They also examined the options for developing new agricultural crops, chiefly, but not only, citrus [33].

The increase in vineyards can especially be seen in the 1943–1945 period, even though the precipitation in these years was slightly below average, which could have potentially reduced the vineyard areas [26]. One explanation for the increase in vineyard areas was that, during the first years of the Mandate in Palestine, the British encouraged the promotion of native crops such as vines [33]. A second reason for the increase in vineyards was the introduction of wine varieties by Jewish settlers and the British [23]. Lastly, the British decided to promote the development of additional water sources for agriculture [33]. Under the civil regime (1920–1948), the British promoted training services to encourage the cultivation of olives and vines, and provided irrigation services [33]. This could explain the planting of vineyards in areas that, before 1943–1945, would have been less hospitable for these plantations. For example, in Hebron in 1943–1945, vineyards were found in the south-east side that were previously uncultivated (Figure 3). Another interesting point to note is that while in the 1873–1874 and 1917 periods vineyards were located in one or two distinct areas, in 1943–1945, vineyards are scattered all over, in large and small areas. This is especially visible in the Ein Karem area (Figure 3). One explanation might be that with the encouragement of agriculture, guidance, and help from the British, and the realization that a larger market for vine products existed, many small settlements decided to plant vineyards as one of their many crops.

While in Palestine until 1917, vineyards were mostly grown for products other than wine, in the Mediterranean region, wine was their main product. Until the middle of the nineteenth century, when French vineyards were attacked by the phylloxera insect, global wine production was dominated by France. From 1880 onwards, France started to import wine from Spain, Italy, and its North African colonies which all saw an increase in the plantation of vines [59]. Moreover, Greece, Spain and Turkey were the main producers of raisins, which were used mostly for wine production as well as in other industries [60]. In 1890, France began to recover and reproduce wine, dictate tariffs and regulations, which continued into the twentieth century, and as a result wine and raisin production in other countries dropped, which in turn decreased the cultivation of vines [59,61,62]. While these changes occurred in the Mediterranean and European regions, in Palestine vineyard growth was not heavily affected by these events and was mainly a result of an increase in domestic consumption. The export of wine from Palestine only began in 1899, and in that year only 366 kilos were exported [63]. In the beginning of the twentieth century, the numbers started to soar in 1936, during the British Mandate, when the export of wine really rose [64].

### 4.2. Olive Groves—Land-Use Change

Olive groves only appear as a separate crop on the 1943–1945 maps. In the 1873–1874 and 1917 maps, the only categories are vineyards and orchards. Two sources can help us fill the gap regarding olive groves in previous periods. The first source is the PEF map memoirs, which describe the surveyed land in great detail. The memoirs give us hints as to where olive trees could be found. For example, in Hebron it is stated that "the hill above Hebron is terraced with stone walls and olive plantations" [41] (p. 307) (Figure 4A). Moreover, "vineyards of Hebron extend over above 6 miles, and olives are also grown there" [41] (p. 297). The description of Bethlehem mentions that "the valleys on the north and south are deep, the sides carefully terraced, vines and olives, figs and either trees are grown along the slopes" [41] (p. 28) (Figure 4B). These descriptions of the PEF memoirs, indicate that other crops existed apart from olive groves, "pomegranates, figs, quinces, and apricots" could be found around Hebron [41] (p. 308). While these sources clearly show that orchards on the maps also included olive groves, they do not specify the amount or size of these areas. Another way to tackle this issue is by examining where olive groves appear on the 1943–1945 maps and whether these exact areas were once orchard areas. There is a strong probability that olive trees that were planted and found in an area in the past could

still be found in that same area after many years. Planting fruit trees is usually done with the thought of long-term profit and especially with regard to olive trees. The results of this research allowed us to estimate the percentages of olive groves (Tables 2 and 3). The assumption according to the results is that Bethlehem had the largest olive groves in the past and still has the largest area in 1943–1945.

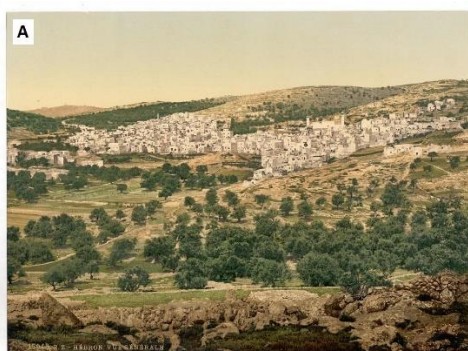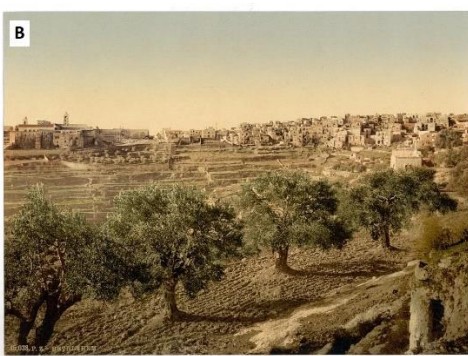

**Figure 4.** Olive groves surroundings the city of Hebron in a photo from 1890–1900 (**A**) [65]. Olive trees in the front of the photo showing the city of Bethlehem from 1890–1900 (**B**) [66].

Among the three research areas, Bethlehem and Ein Karem had the best physical conditions for olive groves in respect to elevation, precipitation, and soil [67,68]. For example, both Bethlehem and Ein Karem are located in the optimal elevation for olive groves (400–800 m), unlike Hebron, which exceeds the optimal elevation (927 m) [68]. Indeed the PEF memoirs also argue that in Hebron "the olive does not flourish well in any part within this Sheet, but the villages in the low hills have a few" [41] (p. 319). While both Ein Karem and Bethlehem had optimal physical conditions for growing olive groves, Bethlehem also had other reasons to have more olive groves. In 1881, Bethlehem was a large settlement with a population of 5000, while Ein Karem was a very small village with a population of only 600 [41]. This could partly explain the smaller amount of olive groves in Ein Karem. However, the stronger explanation is that, from the seventeenth century onwards, Bethlehem's olive trees, unlike other parts in Palestine, were mainly used to produce handcrafted religious souvenirs which would be later sold in Jerusalem or shipped abroad [69]. The Bethlehem orphanage, which was established in 1864, trained the orphanage children to carve in olive wood [69]. This is another indication that a large number of the orchards appearing on the 1873–1974 and 1917 maps were olive groves. Moreover, during the British Mandate, Bethlehem olive-grove areas expanded as olive wood craft continued to flourish in the city [70]. This goes hand in hand with the research results that show that in Bethlehem the size of the olive groves was the largest of all three research areas.

There are two other notably interesting points. The first, has to do with the lack of olive groves east of Bethlehem found across all three research periods. One very logical reason is the fact that the Mediterranean region ends to the east of Bethlehem, and the Irano-Turanian region begins (Figure 1). The Irano-Turanian region does not possess the ideal physical characteristics that both olive trees and vineyards need. Moreover, the east side of the valleys deepen rapidly which make it hard to grow agricultural crops [41]. The second interesting point is that, in all three research areas across all the periods examined, the orchards and olive groves were closer to built-up areas, while the vineyards were located further away from them (Figure 3). The results of the research failed to find a direct link between the distance of vineyards and orchards from built-up areas to the different examined physical factors.

The increase in olive grove areas between 1873 and 1945 found in this research was not a unique phenomenon, but the reasons for it were different. From the beginning of the nineteenth century, all across the Mediterranean region, a movement of Europeans began to encourage the plantation of olive trees in the coastal areas of Northern Africa and the

Levant. This was boosted by the increase in demand for olive oil in France and Italy [71]. In Spain, for example, the increase in olive groves was mostly due to the relative low capital investment needed by the farmers to grow olive trees and the fact that the trees could produces a variety of staple products (olives as food, olive oil, wood for heating, etc.) [72]. In Palestine, olive trees were mainly used in the production of soap with only a small part used to produce edible olive oil [71,73]. In 1885, the export of olive oil reached 916 liters and had peaked in 1890, with 2639 liters of oil but then reduced to a minimum of 32 liters in 1904 [63]. From the end of the nineteenth and the beginning of the twentieth centuries, the increase in olive grove plantations in Palestine was due to the increase in demand for soap—an important source of employment, mostly among the Arab population [71]. In 1885, 0.43 tons of soap were exported, with this export peaking at 4.4 and 5.2 tons in 1895 and 1899, respectively [63]. Moreover, as was previously mentioned, in Bethlehem, olive trees were used to produce souvenirs. From the end of the nineteenth century, with the opening of Palestine to foreign European nations, the tourism industry also increased, as well as the demand for religious handicraft [69,70].

## 5. Conclusions

The goal of the research was to demonstrate the importance of using cartographical sources to obtain a more accurate and complete view of the past. The research examined agricultural land-use changes in three research areas in the Judean region, from the end of the Ottoman Empire to the end of the British Mandate. The research employed different cartographic sources using GIS tools and methods.

The results of this research showed that, in the past, the distribution of vines and olive groves in the Judean region were greatly dependent on physical geographic factors (climate, slopes, direction). The research has shown that Hebron had the optimal physical environmental conditions for vines, and Bethlehem and Ein Karem had the optimal physical conditions for olive trees. Nonetheless, the research also demonstrated that human-related factors also had a large influence on the area size of these land-use categories. For example, although the main olive tree product at that time was designated for the soap industry, in Bethlehem, the hand-crafted religious souvenirs that were made of the wood of olive trees necessitated the growing of olive trees in this area. Despite the ban on wine production during the Ottoman period, Hebron was one of the centers for vines that were grown for products other than wine. Additionally, at the start of the twentieth century, the cultivation of vineyards in Palestine was mainly a result of an increase in domestic consumption, not local-international trade. The research has also demonstrated that, with the passage of time, agricultural land-use area increased and was found in less optimal physical environmental areas, such as the agricultural areas east of Hebron from 1917 onwards. This was brought about by political changes in Palestine, which led to developments and investments in the agricultural sector, mostly during the British Mandate.

While not an aim in and of itself—since this research is founded on historical cartographical sources—it was also critical to verify the level of accuracy and completeness of these sources. One method of examining completeness is by comparing the cartographical source to other cartographical sources from the same place and period [38]. However, we rarely have two different historical sources from the same place and time. A second method is to use a combination of historical sources such as diaries, memoirs, route notes, and travel maps from that same period, or different cartographical sources from the same place but from other periods and examine whether the features appearing on them make sense. This research used a combination of sources to strengthen the level of completeness of the 1873–1874 PEF map. The PEF memoirs, which describe the different regions and provide details on the area's demography, geography, and landscape, were used to check the accuracy of the map. These descriptions helped to complete the overall picture illustrated by the map. Furthermore, two scenery photographs of two of the research areas from 1890–1900 were added to this research to add further reference, albeit general, to the data on the map. Lastly, this research also examined other maps from later periods, 1917

and 1943–1945, to further strengthen our understanding of the past (1873–1874) and of long-term land-use changes.

The introduction of GIS into historical-geographic research has shed light both on the importance of cartographical material and on the large amount of information found in these sources. Lastly, this research has proved that one good way of painting a clearer and broader picture of the past is through a combination of different sources.

**Funding:** This research received no external funding.

**Institutional Review Board Statement:** Not applicable.

**Informed Consent Statement:** Not applicable.

**Data Availability Statement:** Data available on request.

**Acknowledgments:** This research could not have been done without the kind help and the open and free accessibility to use cartographic materials from the following institutions: The Tel-Hai Collage Library Cartographic Collection and the Israel National Library—Eran Laor Cartographic Collection. Thank you.

**Conflicts of Interest:** The author declares no conflict of interest.

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
