# Peer review of "Agricultural Land-Use Changes in the Judean Region from the End of the Ottoman Empire to the End of the British Mandate: A Spatial Analysis"

_ijgi, doi:10.3390/ijgi10050319_

Round 1

Reviewer 1 Report

I read the paper carefully and with interest. The manuscript is well written, but I have some concerns and questions, which has to be considered.

Mayor comments:

The paper focus on three periods. The reasons behind the chosen periods should be explained. Currently, the selected periods seems to me chosen by arbitrary. For the reader it is not clear, why the periods from 1945 to present were not investigated.

The author gives many details on the historical background of the study regions but often missed to connect the historical facts with the results of the study. I suggest revising the manuscript in order to skip those historical descriptions, which appear detached from the results of the study. E.g. line 63-67 “Ottoman Empire” and “British Mandate” Why are these details needed in respect of explaining land-use changes? Chapter 4.1.: To get a stronger focus on the aims of the study chapter 4.1. “Background” should be skipped up to the beginning of page 11. Some of the sentences of line 336-345 could be left.

The author presents the topographic parameters (elevation, slope, aspect) of the three study areas. It would be interesting to compare and discuss the values with the mean of those parameters of other agricultural areas or of the whole region. In addition, it would be interesting to analyse if the measured values significantly differ from the means of other areas.

Additional comments:

It would be interesting to show exemplarily how the landscape has changed over time using a small section of one part of the study area.

Do any historic landscape photographs exist from these study areas? These pictures might be very helpful for ground thrusting. Furthermore, what is about aerial pictures from 1940th? Those sources would be very helpful.

Line 33: You could add Landsat satellite images. Landsat is running since 1972.

Line 41-42: Replace one of the three “changes” in this sentence.

Line 90: A reference is needed here for the phytogeographical regions.

Table 1: If these maps are available via www, the reference should be given.

Table 5: One decimal place might be enough.

Line 350-353. How has the temperature changed between the three time scales?

Line 374-377: Here the results of the study could be mentioned again. How do they fit to that development?

Line 429 onwards: Which role did the beginning of the international wine trade play for this development?

Line 471 “ither”

Line 518-520: Do terraces play a role, which might has biased the slope measurements?

Author Response

Dear Reviewer,

I would like to thank you for your time and effort in reviewing my manuscript with useful and constructive comments. I hope that you will find the revised version clearer, organized, and consistent. Below are my answers to your specific comments.

The paper focus on three periods. The reasons behind the chosen periods should be explained. Currently, the selected periods seems to me chosen by arbitrary. For the reader it is not clear, why the periods from 1945 to present were not investigated. ---In response to this well-taken point I have added at the end of the Introduction a short explanation (lines 108-114). The main reason for which I did not examine the period after 1945 was a technical one (this was also mentioned briefly in section 4.3. lines 770 - 777). After 1948 the Survey of Israel did not produce new maps of two of the research areas (Bethlehem and Hebron) since they were under Jordanian occupation. After 1967 the Survey of Israel started to produce maps of these two areas. Nonetheless, from 1967 onwards, the categories on the maps have changed. For example, from the middle of the 1990s maps of the Survey of Israel have only one category for agricultural areas and these include both orchards and agricultural fields. The lack of maps in certain areas in certain periods and the fact that the categories have changed and have become general prevented making a serious comparison with the older maps. Nonetheless, I do think that the 72 years period examined in this research is interesting since it is a period of rapid and profound change.

The author gives many details on the historical background of the study regions but often missed to connect the historical facts with the results of the study. I suggest revising the manuscript in order to skip those historical descriptions, which appear detached from the results of the study. E.g. line 63-67 “Ottoman Empire” and “British Mandate” Why are these details needed in respect of explaining land-use changes? ---I believe that the historical context had also an influence on the land-use changes examined by the research. Moreover, a not well-informed reader could have otherwise difficulties in understanding the Results and Discussion. Accordingly, I have decided to leave them.

Chapter 4.1.: To get a stronger focus on the aims of the study chapter 4.1. “Background” should be skipped up to the beginning of page 11. Some of the sentences of line 336-345 could be left. --- I have moved the “background” part which was found in section 4.1 and incorporated part of it into the Introduction, as suggested (lines 90-105).

The author presents the topographic parameters (elevation, slope, aspect) of the three study areas. It would be interesting to compare and discuss the values with the mean of those parameters of other agricultural areas or of the whole region. In addition, it would be interesting to analyse if the measured values significantly differ from the means of other areas. --- This point could have been interesting indeed. However, the focus of my work was to compare three research areas among themselves. Moreover, with the time limit given to me (14 days to resubmit the manuscript) I could not have the necessary time to go in the necessary depth to compare the values of the examined parameters to other areas in Palestine at the same period. Comparing other 2-3 areas to the three areas of this research would require at least 3-6 extra months of work. Nonetheless, I have now added a short comparison with vineyards and olive changes in the Mediterranean region in the same period in the Discussion (section 4.1, lines 592-611  and section 4.2, lines 704-722). 

Additional comments:

Do any historic landscape photographs exist from these study areas? These pictures might be very helpful for ground thrusting. Furthermore, what is about aerial pictures from 1940th? Those sources would be very helpful. --- This is indeed a well-taken point. At the beginning of the research, I did try to incorporate aerial photos from 1917 onwards in this research as well as Landsat imagery. The main challenge of using these sources was that interpret aerial and satellite imagery into land-use categories is a completely different process than digitizing categories from historical maps. My main aim in this research was to digitize, categorize and quantify different agricultural land-uses, especially the olive groves and vineyards. This was impossible to do with aerial and satellite imagery. Identification of the land-use categories (vineyards, olive groves) was almost impossible to do by examining these materials.  In addition, the aerial photographs did not cover all the study areas, and this was another reason not to use them. Nonetheless, as you have suggested I have added in this manuscript two landscape photos of olive groves in Hebron and Bethlehem which did not have copyright issues, to strengthen the reliability of the historical maps and data extracted from them (between lines 654-655). I could not find any landscape photo of vineyards. One possible reason is that most landscape photos from that period were of built-up areas. Olive groves were found mostly next to the built-up areas unlike vineyards located further away (this is also a finding that was found in this research).

Line 33: You could add Landsat satellite images. Landsat is running since 1972. --- Please see my answer regarding this suggestion above.

Line 41-42: Replace one of the three “changes” in this sentence. --- I have now rewritten the sentence and delete one of the words “changes” (lines 47-49).

Line 90: A reference is needed here for the phytogeographical regions. --- The map was made by me using the 1984 Floristic Regions of the Land of Israel map. I have now added in the caption (lines 153-154), a reference to the Floristic Regions map (lines 168-169).

Table 1: If these maps are available via www, the reference should be given. --- Most of the maps are found online but to access them you need to be signed in through an institution or organization. I have added in the reference list the internet link of the one map (the PEF map) that is accessible without the need to sign in.

Table 5: One decimal place might be enough. --- I have changed all the numbers in Tables 1, 2, 3 and 5 to one decimal as well as all the numbers that appear in the results section.

Line 350-353. How has the temperature changed between the three time scales?  --- Data on average temperature is available from 1901 onwards, there are no signs of changes between 1901 and 1945. I have also checked the average rainfall. The data is available from 1845 onwards. The average rainfall in the last 150 years is 557mm/year. While some years have seen more rain than others the year 1873 and 1917 had above-average rainfall 1943-1945 had a lower average rainfall. The decline in rainfall between 1943-1945 could have potentially reduced the vineyards more than the olive groves which are less affected by fluctuations of rainfall. Nonetheless, the results do not show a decline but rather an increase of vineyards and olive trees between 1917 and 1943-1945. This is an important point and I have added it in the Discussion (section 4.1, lines 447 and 480-482).

Line 374-377: Here the results of the study could be mentioned again. How do they fit to that development? --- As suggested, I have mentioned the results of 1917 which show a reduction in orchards and vineyards areas from the previous period (lines 472-474).

Line 429 onwards: Which role did the beginning of the international wine trade play for this development? --- I have added a paragraph at the end of section 4.1 regarding the wine trade in the Mediterranean and Europe during the period of this research. I have stated that the international wine trade had little effect on the production of wine in Palestine in this period (lines 592-611).

Line 471 “ither” --- The word was corrected into either (line 638).

Line 518-520: Do terraces play a role, which might has biased the slope measurements? --- The traditional way of building terraces did not create a significant topography transformation of the landscape. Until the middle of the 20th century, most of the terraces were not build at a level platform which would have required a lot of cutting and felling. 

Sincerely,

Reviewer 2 Report

see attached pdf

Author Response

Dear Reviewer,

I would like to thank you for your time and effort in reviewing my manuscript with useful and constructive comments. I hope that you will find the revised version clearer, organized, and consistent. Below are my answers to your specific comments.

General

This is an interesting research topic, a great geographic data set and a relevant result. However, the GIS operations used are rather basic. The historic-geographic description content is phrased as “Discussion” and inappropriate on three accounts. A portion of the “Discussion” is in fact an Intro/Area description that could and should be shortened and transferred to the Intro/Area description. Large portions of the “Discussion” are not a discussion of your Results but a historic-geographic narrative not explicitly related to your findings. Moreover, the Discussions section lacks a comparison with the dynamics of olive groves and vineyards around the Mediterranean in the same period. A systematic literature review would help. Two suggestions for olive groves:

Amate, J.I. 2013. Who boosted olive trees? Nineteenth century olive expansion as a strategy for peasant production (1750-1930). Historia Social 76:25-44.

Amate, J.I. 2012. The nature of Olive expansion in Southern Spain. Historia Agraria 52:39-72. --- I have now added in the Discussion at the end of section 4.1(lines 592-611) a short comparison with vineyards changes in the Mediterranean region in the same period and the same with olive groves at the end of section 4.2. (lines 704-722). Thank you very much for the two references you have suggested, they were very interesting and useful.

The Introduction and Methods section need only minor revisions. The Results and Discussion section have fundamental flaws.

Language copy edits --- Thank you for this main comment regarding English errors. The manuscript was given to an English native person for editing. Moreover, I have corrected the following errors you have mentioned below.  

The title reads as if the changes are traditional --- I have removed the word “traditional” from the title. Now the title of the paper is: “Agricultural land-use changes in the Judean region from the end of the Ottoman Empire to the end of the British Mandate: A spatial analysis” (lines 1-4).

10: Cartographical instead of visual --- As suggested, I have changed the word “visual” into the word “cartographical” (line 10).

12/14 and 58-60: “examins” (2x) is not a goal. Rephrase. Identify? Analyse? --- I have changed this according to your suggestions (line12-13 and 118-120).

17:..in their quantity…is incorrect English --- I have modified the sentence (line 17).

Figure 1: avoid duplication of legend and caption text; delete “Region” in Phytopraphy Reiong --- I have updated Figure 1 – In the legend, I have deleted the word “Regions” from Phytography (lines 146-147).

Table 1: Delete “Publsuh” in the head of column 3 or replace by Publication --- I have updated the title of column 3 into “Year of Publication” (lines 177-178).

Figure 2: avoid duplication of legend and caption text --- I have changed the caption in all three figures (1, 2, and 3) accordingly.

Table 2: reorganize messy texts in the head of the Table --- The text of the table was shortened, and the table was reorganized (lines 276-277).

Table 3: Caption and 204: this is not a statistical “correlation”; maybe spatial correlation, continuity of location --- I have changed the word “correlation” in the Caption into “continuity of location” (line 298). Moreover, I have changed the word “correlation” in previous line 204 into “continuity of location conversion” (line 279).

Table 3: “transformation” may be replaced by conversion --- The word “transformation” has been changed into “conversion” (line 298).

62: ”land” is ambiguous here --- This sentence was deleted.

79: “important” and “ancient” need specification and references. If unavailable delete. --- I have deleted the words “important” and “ancient”.

95: correct separator error in the scale --- It was corrected, thank you (line 159).

100: military maps instead of “war” maps? --- The word “war” was changed into “military” (line 164).

104: a “physical” map is non-sense; physiographic? --- Yes, this was a mistake. I meant a physiographic map. I have now changed the word into “physiographical map” (line 168).

105: : “largely” should be widely. --- The word “largely” was changed into “widely” (line 107).

Results

Figure 2 is a wonderful HGIS output and the best presentation of the findings. However, the figure is not described in the text of the Results section. It deserves a few lines in this section as well as in the Discussion section. --- Thank you, I have added a few lines describing the most important aspect that can be noticed from Figure 2 in the Result (lines 255 – 161).

The Result text (203-219) repeats/duplicates too much of the numerical values presented in Tables 2, 3 and 4. The text should be more general using words such as more, less, similar…….The reader can  look up the numbers in the Table. When two villages show similar values, mention these in the same sentence. Similarly with trends across two villages. Highlight the major findings in words at some level of abstraction in this section. --- As suggested the entire Result section corrected and shortened accordingly (lines 230 – 335).

246: “higher” should be: steeper --- I have changed the word “higher” into the word “steeper” in all the places where it appeared (lines 489-490).

Discussion

The lines 274-314 are not a Discussion of your findings. Delete here. A few key facts may be transferred to the Intro. --- I have moved part of lines and incorporated them into the Introduction of the manuscript, in section 1, as suggested.

There are two subsection 1.4.

The lines 315-345 contain some facts (e.g. population growth; modernization, although that term should be specified in the Intro; export during war; agrarian development efforts by the British) that potentially could be used in an explicit discussion of your findings. The reason for mentioning Muslims versus Christians is unclear and should be deleted or brought in an explicit relation to the discussed findings on olive groves. Further down in the Discussion, the distinction between Christian Arabs and Jews versus Muslim Arabs may be relevant in developments in vineyard trends. For clarity sake you should use Arab Muslims were applicable (wine grapes). ---Firstly, I have specified in a few words what is the meaning of “modernization” in Palestine– construction of railroads, the start of export of goods, and growth in population (lines 97-100). Secondly, I have mentioned in some places in the manuscript Christian Arabs since this is relevant regarding the wine production issue (which was banned for Arab Muslims). I have now explained it furthermore in the Discussion. In the rest of the manuscript, I use just the word Arabs since their religion is irrelevant.

380: “figure 2” is referred to , but is not included. --- Figure 2 is found in the Result section. Previously it was written “Figure 3” and I have corrected it into “Figure 2”.

449-459: is not related to a Discussion of your findings. Delete. --- As it was written, I do agree that this part seemed unrelated to the discussion. I have now rewritten the discussion incorporating just some of this information since I believe it is relevant.

505-510: is a Result. Transfer to the appropriate section. --- As suggested, I have transferred these lines to the Result section.

Sincerely,

Reviewer 3 Report

This is very interesting paper that is very well written. The paper reads easily and quite well. I was especially happy with the history and context for the research. While the GIS analyses undertaken were not extensive, these were quite appropriately employed for a study of this nature. Except for a few minor corrections, this work is quite acceptable. 

The following needs to be checked -

the number 8,94 at the end of line 174. 

replace 'were' with 'where' at the start of line 283

replace 'remined' with remained on line 552

Remove book after memoir in lines 543 and 563.  

Author Response

Dear Reviewer,

I would like to thank you for your time and effort in reviewing my manuscript. Thank you very much for the comments. The resubmitted manuscript was given to a native English professional for editing. Moreover, I have corrected the following errors you have mentioned below.

This is very interesting paper that is very well written. The paper reads easily and quite well. I was especially happy with the history and context for the research. While the GIS analyses undertaken were not extensive, these were quite appropriately employed for a study of this nature. Except for a few minor corrections, this work is quite acceptable.

The following needs to be checked.

the number 8,94 at the end of line 174. --- I have shortened the Result section. This number was deleted.

replace 'were' with 'where' at the start of line 283 --- The word was corrected into “where”. The entire section was moved to the Introduction (line 51).

replace 'remined' with remained on line 552 --- The word was corrected to “remained” (line 754)

Remove book after memoir in lines 543 and 563.  --- The word “book” was deleted from both places as well as from other places in the manuscript (i.e. lines 476, 484, 632-633, 640, etc.)

Sincerely yours,